# The Quality and Bacterial Community Changes in Freshwater Crawfish Stored at 4 °C in Vacuum Packaging

**DOI:** 10.3390/molecules27238618

**Published:** 2022-12-06

**Authors:** Liang Qiu, Yunchun Zhao, Hui Ma, Xiaofei Tian, Chan Bai, Tao Liao

**Affiliations:** 1Institute of Agro-Products Processing and Nuclear Agricultural Technology, Hubei Academy of Agricultural Sciences, 5th Nanhu Aevenue, Wuhan 430064, China; 2Key Laboratory of Cold Chain Logistics for Agro-Product, Ministry of Agriculture and Rural Affairs, Wuhan 430064, China; 3Guangdong Key Laboratory of Fermentation & Enzyme Engineering, School of Biology and Biological Engineering, South China University of Technology, 382 East Out Loop, University Park, Guangzhou 510006, China

**Keywords:** crawfish, food preservation, microbial diversity, volatile flavor compounds function prediction

## Abstract

Crawfish can be easily spoiled due to their rich nutrition and high water content, which is difficult to preserve. In this study, the dominant spoilage organisms in crawfish which were stored at 4 °C in vacuum packaging were identified by high-throughput sequencing technology; after sequencing the full-length 16S rRNA gene, the changes in the bacterial community structure, diversity and quality (texture, flavor, etc.) were analyzed. Our results reflected that the specific spoilage organisms (SSOs) of crawfish were *Aeromonas sobria*, Shewanella putrefaciens, Trichococcus pasteurii and *Enterococcus aquimarinus*, since their abundances significantly increased after being stored for 12 days at 4 °C under vacuum conditions. At the same time, the abundance and diversity of the microbial community decreased with storage time, which was related to the rapid growth of the dominant spoilage organisms and the inhibition of other kinds of microorganisms at the end of the spoilage stage. Function prediction results showed that the gene which contributed to metabolism influenced the spoilage process. Moreover, the decline in texture of crawfish was negatively correlated to the richness of SSOs; this may be because SSOs can produce alkaline proteases to degrade the myofibrillar protein. On the contrary, the unpleasant flavor of crawfish, resulting from volatile flavor compounds such as S-containing compounds and APEOs, etc., is negatively correlated to the richness of SSOs, due to the metabolism of SSOs by secondary metabolites such as terpenoids, polyketides and lips, which can lead to decarboxylation, deamination and enzymatic oxidation. These results are very important to achieve the purpose of targeted inhibition of crawfish spoilage at 4 °C in vacuum packaging.

## 1. Introduction

For many countries, aquaculture, especially of crustaceans such as crawfish, is an essential economic pillar industry and is an increasingly important component in the global food supply [1]. In 2020, the Food and Aquaculture Organization of the United Nations announced that the yield of freshwater crawfish reached 0.35 billion tons. Since it has a high protein content and low fat and cholesterol content, its industry has had a huge economic effect, especially for China (CNY 344.8 billion in 2020) [2]. Therefore, the spoilage and preservation of crawfish has attracted scientific attention to extend the shelf life of crawfish and maintain its nutritional composition during transportation and storage [3].

The main reason behind the spoilage of aquatic products is microbial activity; the enzymes which are involved in metabolic function can decompose the protein into small molecular compounds such as nitrogen- and sulfur-containing compounds [4]. These compounds will cause aquatic products to emanate bad smells, prompting the sensory rejection of consumers [5]. During the period of food decay, aquatic products display extremely rich microbial communities, but only a few microorganisms contribute to the process of food spoilage. These microorganisms are called specific spoilage organisms (SSOs) [6]. The key factor to understanding the spoilage mechanism is to reveal changes in the microbial community. With the rapid development of high-throughput sequencing (HTS) technologies, many SSOs have been found by researchers [7]. For example, H. Wang et al. discovered that some bacteria, such as *Pseudomonas fluorescens*, *Aeromonas veronii* and *Shewanella putrefaciens*, were less abundant in fresh samples, but at the end of storage, they were the main reason behind grass carp spoilage [8]. The same results were also found by R.E. Levin, who found that the specific spoilage organisms were closely related to flavor substances which brought fish to sensory rejection point [9]. However, reports about microbial changes in crawfish are very few and not systematic [10]. The species of specific spoilage bacteria in aquatic products processed by different ways are also different. Research has shown that the main microorganism species in fresh crawfish include *Flavobacteria*, *Aeromonas*, *Shewanella*, *Microbacter* and *Bacillus* [11], while the main species in cooked flavored crawfish was *Bacillus* [12].

Moreover, different temperatures will cause different SSOs. For example, *Staphylococcus* had the strongest ability to cause putrefaction when halogenated crawfish was stored at −4 °C, while *Aeromonas jandaei* was the main SSO at room temperature [13]. Crawfish is a seasonal aquatic product, and it is mainly fished from April to July. To extend the storage time and maintain quality, freezing, refrigeration and vacuum packaging are used to store crawfish [14]. Among those methods, refrigeration and vacuum packaging are the most common methods due to the fact that freezing easily leads to dehydration of muscles and a decrease in quality [15]. Thus, understanding the microbial structure and diversity of crawfish at 4 °C in vacuum packaging and studying the specific spoilage bacteria causing product deterioration is very important to achieve the purpose of targeted inhibition of product spoilage [16].

Therefore, the objective of this study was to analyze the structure (species level) and composition of the microbial community in crawfish during storage at 4 °C by using third-generation sequencing (TGS) amplicons. In addition, other indicators such as total volatile base nitrogen (TVB-N) and total aerobic microorganisms, texture and volatile flavor compounds were also determined to understand crawfish quality changes at 4 °C.

## 2. Materials and Methods

### 2.1. Materials

Live crawfish with an average weight of 30 ± 2.4 g and average length of 20 ± 1.8 cm were purchased in June from Qiyimeng fresh market, Wuhan. The edible part of the crawfish is mainly the shrimp tail, so the slaughter process was as follows: once brought to the laboratory, crawfish were cleaned in water, the head and sand vein were removed, the tail was rinsed and then the crawfish were quickly vacuum packed. Ten crawfish (randomly picked) were set as one batch and samples were stored at 4 °C for further analysis. The initial crawfish was marked as C-0-D0 and the crawfish which were stored at 4 °C in vacuum packaging for 12 days were marked as C-0-D12, respectively. Temperature was controlled by ice box (BCD-556WKGM, Meidi, Foshan, China).

Plate count agar (PCA), boric acid, MgO and other reagents are purchased from Sinopharm Chemical Reagent Co., Shanghai, China. DNA Extraction kit, DNA polymerase were purchased from McLean Biochemical Technology Co., Ltd., Shanghai, China. All the reagents were of analytical grade.

### 2.2. Determination of the Total Volatile Base Nitrogen (TVB-N) and Total Aerobic Microorganisms

TVB-N value was estimated by the FOSS method. Crawfish were hacked through the packaging, cleaned and drained in sterile environment (5.0 g) and homogenized with MgO (0.5 g) by a mixer. At last, all the mixture was inserted into FOSS digestion tube. TVB-N value was measured by Automatic Kjeldahl Apparatus (8400 Kjeltec Distillation, FOSS Analytical Instrument Co., Ltd., Shanghai, China), and the levels of TVB-N were expressed as mg N/100 g of crawfish [14].

The total aerobic microorganism value was estimated as follows: crawfish were hacked through the packaging, cleaned and drained in sterile environment (10 g) and homogenized, then put into a sterile bag with 90 mL aseptic sterile saline, then beat by a beating homogenizer for 3 min to make a uniform bacterial suspension. The bacterial suspension was diluted by 10× dilution method. The appropriate bacterial solution (0.1 mL) with three gradients was coated on the surface of plate. The total aerobic microorganism value of the dilution was measured using the standard plate count method using nutrient agar after incubating at 30 °C for 48 h and the value of total aerobic microorganisms were expressed as log CFU/g of crawfish [14].

### 2.3. Texture Profile Analysis (TPA)

Crawfish were hacked through the packaging, cleaned and drained in sterile environment (10 g) and homogenized. TPA was evaluated by hardness, springiness, chewiness and resilience using a Lab Pro texturometer (Food Technology Corp., Sterling, VA, USA). The size of flat-bottom cylindrical probe was 50 mm (diameter) and the experimental conditions were as follows: initial strength, 0.8 N; probe recovery height, 25 mm; test speed, 20 cm/min; and sample compression rate, 30%. Each experiment was replicated 3 times.

### 2.4. GC-MS Measurement of Volatile Flavor Compounds

Crawfish were hacked through the packaging, cleaned and drained in sterile environment (8 g) and homogenized. Headspace solid phase microextraction conditions were: putting 8 g samples into HS-SPME sample flask, adding 4~5 mL 25 g/100 mL of sodium chloride solution, water bath for 10 min at 65 °C, inserting a 75 μm CAR/PDMS extraction head (after aging treatment) into the headspace flask, extracting for 40 min at 65 °C and then quickly placing the extraction head into the 250 °C injection port to desorb for 3 min. Chromatographic conditions were as follows: DB-5MS capillary column (30 m × 25 mm, 0.25 μm); carrier gas: high-purity helium; helium gas flow rate: 1.0 mL/min; injection volume: 1 μL; splitless sampling; inlet temperature: 250 °C; temperature programming: column temperature of 40 °C for 2 min, increasing the temperature to 60 °C at a speed of 3 °C/min and maintaining for 5 min, then increasing the temperature to 100 °C at a speed of 4 °C/min and then to 240 °C at a speed of 8 °C/min and maintaining for 10 min. Mass spectrometry conditions were as follows: ion source temperature: 230 °C; electron energy: 70 eV; ionization mode: electron bombardment of ion source; mass scan range: *m*/*z* 40~400.

### 2.5. DNA Extraction

The total DNA of crawfish was extracted by cetyl trimethyl ammonium bromide (CTAB). Once the DNA was extracted, we used 1% agarose gels (Life Technologies, Grand Island, NY, USA) to quantify its concentration and purity. At last, we used sterile water (1 ng/μL) to dilute the DNA [17].

### 2.6. 16S rDNA Amplification and SMRT Sequencing

In this study, PCR was used to amplify the 16S rDNA of the microorganisms; the primers were 27F (AGAGTTTGATCMTGGCTCAG) and 1492 R (ACCTTGTTACGACTT). The ends of 5′ primer used paired 16-nt symmetric barcodes. The process of thermocycling was: first, using 95 °C in 5 min for 1 cycle and using 95 °C in 1 min for 28 cycles; then, using 58 °C for 1 min, and 72 °C for 2 min; at last, 1 cycle for 72 °C in 10 min. The amplicon quantification used Agilent DNA 1000 Kit and an Agilent 2100 Bioanalyzer (Agilent Technologies). The DNA libraries used amplicons (2 μg/sample) by Pacific Biosciences Template Prep Kit 2.0 [18,19].

### 2.7. Sequence Analyses

The raw sequencing data were analyzed by the RS_ReadsOfinsert.1 (available under the SMRT Portal, version 2.7) [20]. Before extracting high-quality sequences, the barcode sequences were removed by Quantitative Insights Into Microbial Ecology package (QIIME; version1.7). Then, we used PyNAST (100% clustering of sequence identity) and UCLUST to align the most abundant sequence of each cluster. Afterward, we used UCLUST (cut-off at 98.65%) to change the unique sequences to OTUs. Next, chimeric OTU sequences were screened and removed by ChimeraSlayer [21]. At last, we used Ribosomal Database Project (RDP) II and Greengenes (version 13_8) databases (minimum bootstrap threshold of 80%) to assign the remaining OTUs taxonomically [22].

The de novo taxonomic tree was built by the representative chimera-checked OTU dataset using FastTree and the alpha diversity was calculated [23]. All samples’ rarefaction curves and rank abundance curves were constructed to assess the sequencing depth and biodiversity richness. After calculating the Shannon diversity index, a Venn diagram was created by Visual Paradigm Online [24].

### 2.8. Statistical Analysis

All experiments were performed in triplicate and the results are expressed as the mean ± standard deviation (SD). SPSS Statistics 20.0 (IBM SPSS Statistics, Ehningen, Germany) was used for data statistical analysis, such as TVB-N, TPC and TPA et al.; the alpha diversity and one-way ANOVA were calculated; and the value of *p* < 0.05 was used to indicate significant deviation.

## 3. Results and Discussion

### 3.1. Total Volatile Base Nitrogen (TVB-N) and Total Aerobic Microorganisms

TVB-N was mainly contributed by ammonia which was produced from bacterial catabolism. From the hygienic standards for marine products, the acceptable TVB-N values for marine fish and shrimp should be no more than 20 mg N/100 g [25]. Our results (Figure 1a) reflected that the initial TVB-N value of crawfish was 5.41 ± 0.26 mg N/ 100 g and it exceeded 20 mg N/100 g on the 12th day. Meanwhile, the results of total aerobic microorganisms (Figure 1b) showed that on the 12th day, the total aerobic microorganism values of crawfish was higher than 6 log CFU/g, which means the crawfish were incredibly spoiled [26]. Thus, the quality guarantee period of crawfish is 12 days in our study. Some studies showed that crawfish will become spoiled after 6 days at 4 °C; this may be because crawfish were in vacuum packaging in this study and only facultative anaerobic bacteria contributed to the spoilage process.

### 3.2. Texture Profile Analysis (TPA)

The TPA results are shown in Figure 2, including hardness (a), springiness (b), chewiness (c) and resilience (d). With the extension of storage time, the value of hardness decreased faster and other indexes had the same results. This may be because the texture of crawfish could be related to the diameter of the muscle fibers, the content of free water and myofibril attachments [27]. During the early stage, low temperature could control water vapor spreading from the crawfish muscle and improve their water-holding capacity [28]. With longer time, the spoilage bacteria increased rapidly. The myofibrillar protein was degraded and muscle fiber bundles were seriously damaged; muscle fibers were broken and loose, resulting in a decrease in hardness, springiness, etc. [29].

### 3.3. Volatile Flavor Compounds

As shown in Figure 3, in total, 81 types of volatile compounds were detected from the crawfish, which were alkanes (4 types), alcohols (5 types), acids (5 types), aldehydes (2 types), esters (9 types), benzenes (17 types), terpenoids (27 types), N-containing compounds (isopropylamine hydrochloride and dimethylamine), S-containing compounds (dimethyl trisulfide and tetrasulfide, dimethyl and alkyl phenol polyoxyethylene ether) and APEOs (alkylphenolethoxylates) (2-(Ethenyloxy)-ethnolect). Among those compounds, benzenes and terpenoids increased the most; this may be because they are mainly produced from fat oxidation or PHE (phenylalanine) catabolism [30]. Meanwhile, those compounds which presented a negative impact on the flavor of aquatic products, such as N-containing and S-containing compounds and APEOs, also increased, which means the products were spoiling [31]. The decreased compounds mainly included alkanes, aldehydes, etc. With a longer period, they can be transformed into other compounds, such as benzene, etc., and change the flavor of crawfish [32,33,34,35].

### 3.4. Dataset Features and Species Richness

A total of 14,892 and 19,965 16S rDNA raw reads were generated from C-0-D0 and C-0-D12, respectively (Appendix A). In addition, we used PyNAST alignment and identity clustering to compartmentalize 14,466 and 19,802 reads of each sample.

The bacterial community changes of crawfish are shown in Table 1 (Top 10). Through these results, we could find the changing trend in species at different environments, evaluate the species with the most significant or most stable change, the dominant species, species annotation rate, etc., and the biological relationship between species and groups. The bacterial community of C-0-D0 included *Paucibacter oligotrophus*, *Cloacibacterium rupense*, *Flavobacterium sangjuense*, *Bacteroides nordii*, *Nitrospira moscoviensis*, *Brachymonas chironomi*, etc., while at the end of storage (C-0-D12), the main species were *Aeromonas sobria*, *Shewanella putrefaciens*, *Trichococcus pasteurii* and *Enterococcus aquimarinus*. The changes of the species richness were as follows: *Paucibacter oligotrophus* (20.59%→0), *Cloacibacterium rupense* (15.57%→0), *Flavobacterium sangjuense* (8.18%→0) almost disappeared and *Aeromonas sobria* (0.03%→55.092%), *Shewanella putrefaciens* (0.57%→19.29%), *Trichococcus pasteurii* (0.03%→13.99%) increased obviously in C-0-D12. Moreover, we found a new bacterium, *Enterococcus aquimarinus*, in C-0-D12 (0%→10.39%). This may be because *Enterococcus aquimarinus* had too low abundance in C-0-D0 to be detected.

The reason that the initial species richness decreased may be because (1) in this study, crawfish were vacuum packaged at 4 °C, and the SSOs that we detected belong to psychrophilic and facultative bacteria [36,37,38]. Anaerobic and low-temperature environments provide favorable conditions for the growth of SSOs. (2) During the storage time, SSOs increased and released quorum sensing (QS) molecules such as Luxl (AHL synthase) and DKPs (diketone piperazines) [39]; these QS signal molecules can promote the related spoilage phenotypes and inhibit the growth of competitive bacteria.

At present, the reports about microbial changes in crawfish are very few and not systematic [40]. This study used third-generation sequencing (TGS) amplicons to detect the bacterial community on a species level; compared to other studies (Appendix A) [41,42,43,44,45,46,47], we not only found some common SSOs, such as *Aeromonas sobria*, *Shewanella putrefaciens*, etc., but also found a special bacterium: *Enterococcus aquimarinus*. *Enterococcus* will appear as an SSO if aquatic products come from contaminated water [48]. Crawfish is a benthos; it will be exposed to many pollutants. However, few studies showed that *Enterococcus aquimarinus* is an SSO during crawfish spoilage [49]. Thus, *Enterococcus* should be considered in the study of crawfish spoilage in future.

### 3.5. Alpha Diversity of the Microbiota of the Crawfish

Alpha diversity is often used to evaluate microbiota richness and individual distribution uniformity. With longer time, the Shannon, Simpson and OUT index of crawfish all decreased (Table 2). Moreover, Venn diagram (Figure 4b) results showed that the common species between C-0-D0 and C-0-D12 were 19, while the total species of C-0-D0 were 109 and C-0-D12 were only 26. These results meant that the spoilage bacteria replaced other flora in order to become the main microorganisms.

In terms of microbiota variety, this study’s Shannon index on C-0-D0 was 4.46, and its Simpson index was 0.91, both of which were higher than those of previous research that gave alpha diversity data [50,51]. It shows that using 16S full-length sequencing, crawfish microbiota diversity is larger, and crawfish microbiota are more fully shown.

Furthermore, the rank abundance curve (Figure 5) is also an index for us to evaluate the sequencing results; the rank abundance curve can directly reflect the taxonomic richness and evenness of the samples. The richness and evenness of the microorganisms in the samples were reduced during the preservation procedure, as seen by the rank abundance curve. It also demonstrates that under the vacuum-packed 4 °C preservation environment, SSOs proliferate rapidly, with strains such as *Shewanella putrefaciens* becoming the dominant strain in the system and inhibiting the growth of most bacteria [52].

### 3.6. Correlation Analysis between Dominant Bacteria and Quality

The correlation between dominant bacteria and flavor, TVB-N and texture are shown in Figure 6. The results indicated that SSOs were significantly positively correlated to bad flavor (Figure 6a), with the release of compounds such as benzene compounds, APEOs and S-containing compounds, etc.; TVB-N also had the same results. On the contrary, texture was significantly negatively correlated with SSOs (Figure 6b).

Crawfish is an aquatic product with high protein content; thus, its flavor is mainly related to protein. During storage time, endogenous enzymes lead to the decomposition of muscle protein to produce molecule peptides, amino acids and other intermediates [53]. Meanwhile, the metabolism of SSOs produces various enzymes; these enzymes react with intermediates which come from autolysis through decarboxylation, deamination and enzymatic oxidation [54]. These reactions produce ammonia, amine, aldehyde, alcohol, acid and ester, mercaptan, H_2_S, indole and other volatile products, making the crawfish produce an unpleasant odor and fishy smell [55]. For example, JIA et al. confirmed that ketones (especially C7–C9 ketones) and sulfur compounds were produced by psychrophilic bacteria and *Shiva putrefactoris* during the storage of silver carp meat [56]. The results of function prediction also proved this; the abundance of metabolism which involved terpenoids, polyketides and lips increased in C-0-D12.

The texture of crawfish is mainly correlated to myofibrillar protein and includes actomyosin, actin and myosin, etc. Studies showed that SSOs can release alkaline proteases to degrade myofibrillar protein [57]; for example, Qiaoling Zhao et al. found that during the storage of *Collichthys niveatus*, SSOs such as *Bacillus licheniformis* released alkaline proteases which accelerated the degradation process of myofibril structure protein, destroyed fish tissue structures and led to a decline in fish quality [58]. Thus, the alkaline proteases produced from *Aeromonas sobria*, *Shewanella putrefaciens*, *Trichococcus pasteurii*, etc., can decrease the texture of crawfish [59].

### 3.7. Function Prediction

By functional annotation, OTUs were classified to four levels. In this study, we mainly discuss Levels 1 and 2 since they represent metabolic paths and function. Metabolism can guide microbial interactions. For example, it can induce enzymatic reactions successively carried out by different microbial species [60]. Complementary activity of the metabolism and metabolic pathways of different microorganisms could enhance the spoilage process’ complexity. The genetic information processes and cellular processes could be related to the stress response at low temperature (Appendix A). In total, there were 7 classes (Levels 1) and 27 metabolic functions (Levels 2) inferred in the crawfish samples (Figure 7).

Table 3 shows the different abundance of function between C-0-D0 and C-0-D12 (Top10); they include carbohydrate metabolism, metabolism of cofactors and vitamins, amino acid metabolism and others. These findings indicated that (1) during storage, some carbon and nitrogen source compounds, such as glucose and glucose-6-phosphate, can be utilized by bacterial activities; (2) microbial proteases can hydrolyze the structural protein into peptides and amino acids, which induces changes in the physicochemical properties of crawfish i.e., texture, moisture distribution and color; (3) produced peptides could be transferred into bacterial cells and subsequently degraded into amino acids. The amino acids undergo transamination, deamination and decarboxylation metabolisms in cytostomes, by which α-keto acids, ammonia and various kinds of biogenic amines were produced, respectively [61,62].

The metabolism of terpenoids and polyketides may be related to enzymatic reactions which relate to flavor [37]. Terpenoids and polyketides are secondary metabolites produced by bacteria, fungi, plants and animals. With longer time, the richness of SSOs increased, resulting in more enzymes participating in the reaction. That is the reason why the content of terpenoids increased the result of volatile flavor compounds. Obviously, enzymatic reactions will consume much protein and fat, reducing the quality and shelf life of crawfish.

The unsaturated fatty acid translates to polyunsaturated fatty acid, which will produce a peculiar smell [63]. As the proteins and peptides’ decomposition needs energy, xenobiotic biodegradation may contribute to the ATP metabolism [64]. Additionally, the degradation of nucleotides, especially ATP-related compounds, is highly correlated with the flavor deterioration of fishery products.

In addition, DNA replication and repair and cell motility were also changed in C-0-D12 compared to C-0-D0. DNA replication and repair is a response to environmental factors, since DNA damage responses are elicited more often in extreme environments due to low temperature. The same results were also reported by Abramova et al., who found that the genetic information processes involved in the regulation of DNA repair and replication proteins in the ocular surface microbials could be changed under UV irradiation [65]. Cell motility perhaps contributed to muscle autolysis [66]. During spoilage processes, the muscles of crawfish undergo the process of contraction, putrefaction, autolysis and thawing. The muscle’s free state could promote the cell’s fluidity.

According to the results, the spoilage of crawfish under low temperatures can be divided into two stages [67]. First, from the beginning of storage to sensory rejection, the primary reaction is protein degradation, the conditions of the reaction contributing to ATP and enzyme metabolism. Additionally, the microbial composition undergoes dramatic changes, usually reflecting a decreased microbial richness and diversity. Only a few taxa of bacteria become dominant in the tissue with increased storage time. During the second stage, from sensory rejection to complete spoilage, amino acids translate to various kinds of biogenic amines by SSOs. Thus, crawfish are generally spoiled once the microbial community reaches a stable stage. New methods for preserving crawfish—for instance, using promising bio-preservation materials, compounds and even bacteriophages—could be developed to inhibit the growth of spoilage bacteria with high efficiency and specificity in lysing target bacteria.

## 4. Conclusions

In this study, the crawfish were stored at 4 °C in vacuum packaging. Our results suggested that the preservation temperature is a critical factor in the decay of crawfish and the shelf life of crawfish will extend to 12 days at 4 °C in vacuum packaging. With longer time, the total species of bacteria decrease from 109 to 26 and the SSOs of crawfish at 4 °C in vacuum packaging are *Aeromonas sobria*, *Shewanella putrefaciens*, *Trichococcus pasteurii and Enterococcus aquimarinus*. Moreover, the decline in texture of crawfish is negatively correlated to the richness of SSOs; this may be because SSOs can produce alkaline proteases to degrade the myofibrillar protein. On the contrary, due to the metabolism of SSOs, secondary metabolites such as terpenoids, polyketides and lips can lead to decarboxylation, deamination and enzymatic oxidation, which creates the unpleasant flavor of crawfish from volatile flavor compounds such as S-containing compounds and APEOs, etc., which are also negatively correlated to the richness of SSOs. Functional analysis results suggested that carbohydrate metabolism, metabolism of cofactors and vitamins, amino acid metabolism and metabolism of other amino acids had a strong relationship with microbial decay. The findings of the study were of great significance in the prevention of and reduction in the spoilage and decreased flavor of crawfish.

## Figures and Tables

**Figure 1 molecules-27-08618-f001:**
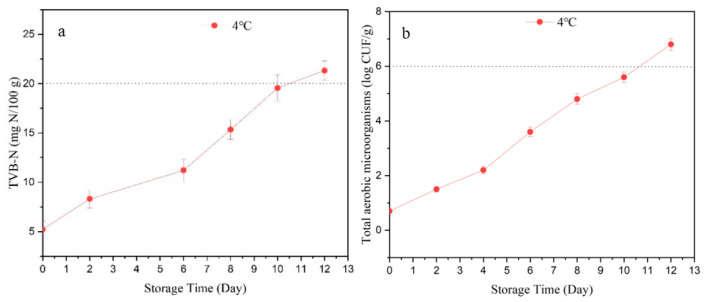
The TVB-N (**a**) and count of aerobic microorganisms (**b**) of crawfish at 4 °C in vacuum packaging.

**Figure 2 molecules-27-08618-f002:**
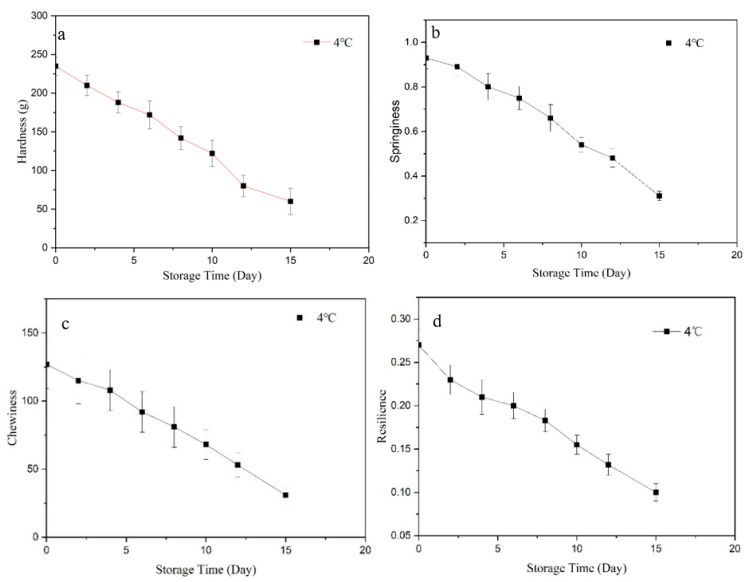
The texture of crawfish at 4 °C in vacuum packaging. ((**a**): hardness; (**b**): springiness; (**c**): chewiness; (**d**): resilience).

**Figure 3 molecules-27-08618-f003:**
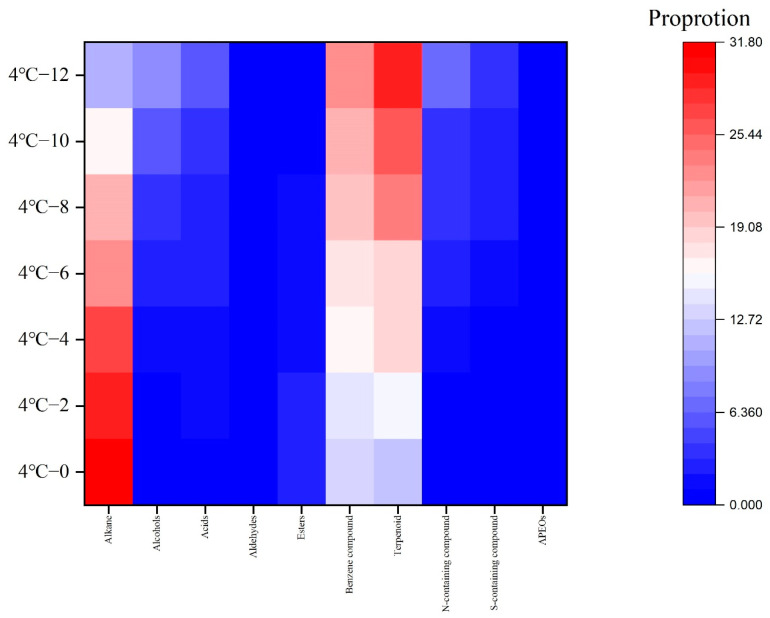
The volatile flavor compounds of crawfish at 4 °C in vacuum packaging.

**Figure 4 molecules-27-08618-f004:**
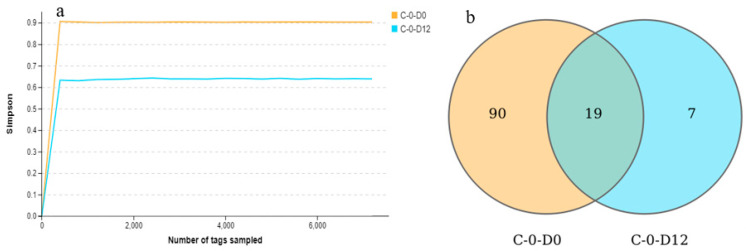
Shannon rarefaction plot (**a**) and species Venn diagram (**b**) of each sample.

**Figure 5 molecules-27-08618-f005:**
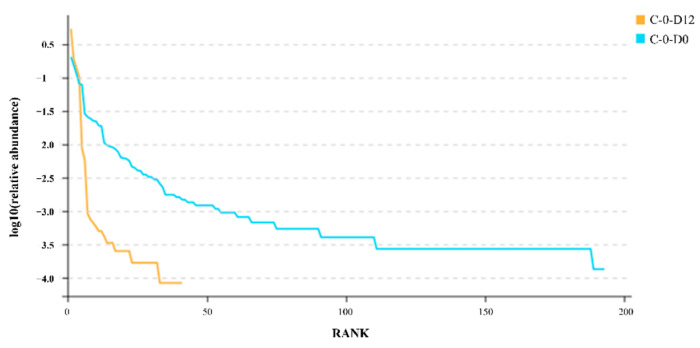
Rank abundance curve.

**Figure 6 molecules-27-08618-f006:**
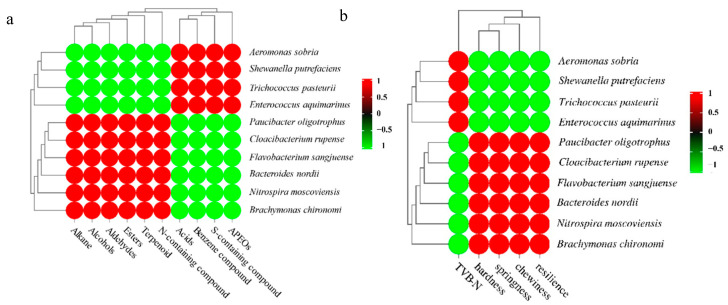
Correlation heatmap of dominant bacteria and quality (volatile flavor compounds (**a**) and TVB-N, texture (**b**)).

**Figure 7 molecules-27-08618-f007:**
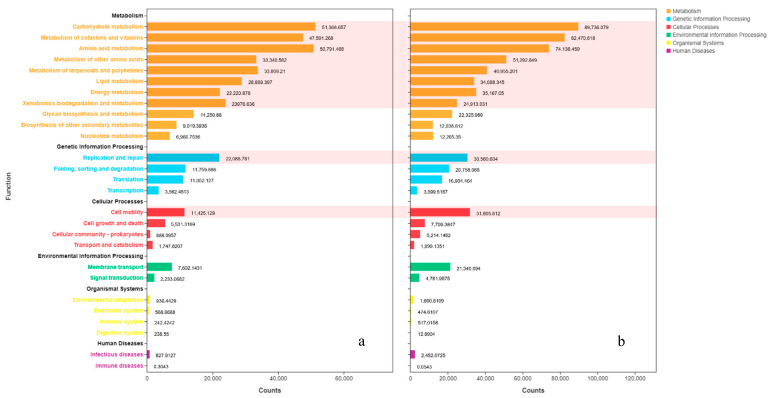
Function distribution bar chart of gene in C-0-D0 (**a**) and C-0-D12 (**b**) (different abundance of function was marked in the background).

**Table 1 molecules-27-08618-t001:** The abundance of species for crawfish stored at 4 °C in vacuum packaging.

Species	C-0-D0 (%)	C-0-D12 (%)
*Aeromonas sobria*	0.03	55.09
*Paucibacter oligotrophus*	20.59	0
*Shewanella putrefaciens*	0.57	19.29
*Cloacibacterium rupense*	15.57	0
*Trichococcus pasteurii*	0.03	13.99
*Enterococcus aquimari* *nus*	0	10.39
*Flavobacterium sangjuense*	8.19	0
*Bacteroides nordii*	7.85	0.02
*Nitrospira moscoviensis*	5.19	0
*Brachymonas chironomi*	2.58	0

**Table 2 molecules-27-08618-t002:** Alpha diversity of bacterial flora for crawfish stored at 4 °C in vacuum package.

Samples	OTUs	Shannon	Simpson
C-0-D0	193	4.46	0.91
C-0-D12	41	1.88	0.64

**Table 3 molecules-27-08618-t003:** The different abundance of function (Top10).

Level 1	Level 2	C-0-D0 (Abundance)	C-0-D12 (Abundance)
Metabolism	Carbohydrate metabolism	51,368.66	89,736.08
Metabolism	Metabolism of cofactors and Vitamins	47,591.27	82,470.62
Metabolism	Amino acid metabolism	50,791.47	74,136.46
Metabolism	Metabolism of other amino acids	33,340.58	51,292.85
Metabolism	Metabolism of terpenoids and polyketides	33,809.21	40,955.21
Metabolism	Lipid metabolism	28,889.41	34,088.35
Metabolism	Energy metabolism	22,220.88	35,167.05
Genetic information	Replication and repair	22,088.78	30,560.60
Metabolism	Xenobiotics biodegradation and metabolism	23,976.63	24,913.03
Cellular processes	Cell motility	11,425.13	31,805.81

## Data Availability

Not applicable.

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
