# Peer review of "The Quality and Bacterial Community Changes in Freshwater Crawfish Stored at 4 °C in Vacuum Packaging"

_molecules, 2022, doi:10.3390/molecules27238618_

Round 1

Reviewer 1 Report (Previous Reviewer 2)

This article deals with the investigation on the quality and bacterial community changes of crawfish stored at 4℃. This MS is interesting, but the design of experiments/ how’s to interpret /discussion the data should be improved. Some of my suggestions are as follow:

P.S. Please add the line number throughout the text during revision stages.

ABSTRACT

- I suggest the outcome/result from the study should more in-depth discussion likes what microbial species/genes contributed/indicated to the spoilage of the fish and how? How’s those changes related with its meat quality (both texture and flavor/odor)? etc.

- Last sentence of the abstract: Author stated that the findings from this study can provide a theoretical guidance for the storage of crawfish at 4℃. How???? Based on microbial community changes??? Or what??? Please clarify this point.

INTRODUCTION
- Some of review literature is not related to the work likes the paragraph of “Different packaging treatments”. However, instead of introduction, I think author should design to do experiment in all vacuum or MAP, compared with control. Unless, author should give the reason why select the vacuum for this study?

MATERIAL AND METHODS

- I wonder why author use vacuum packaging for- this study? I had strong impact to the changes in both microbial population and quality, compared to traditional hand-wrapped or normal packaging. Why should author not design the experiment to compare between vacuum and control or etc.? (Moreover, I suggest the “vacuum packaging” should appear in the topic).

- What is the slaughter process for this study since author start with the alive fish. How it proper for ethics or animal welfare? Please state the process in this part.

RESULT AND DISCUSSION

- I suggest author add more discussion to state the relationship between the changes in bacterial community and the changes in texture/flavor. This should be create the interesting data of this work.

P.S. reference is not appear in the correct format, please corrected it.

Author Response

Dear Editors and Reviewers:

Thank you for your letter and for the reviewers’ comments concerning our manuscript entitled “Bacterial community changes during freshwater-crawfish spoilage under the low- temperature storage” (ID: 1994477). Those comments are all valuable and very helpful for revising and improving our paper, as well as the important guiding significance to our researches. We have studied comments carefully and have made correction which we hope meet with approval. Revised portion are marked in red in the paper. The main corrections in the paper and the responds to the reviewer’s comments are as flowing:

R1:This article deals with the investigation on the quality and bacterial community changes of crawfish stored at 4℃. This MS is interesting, but the design of experiments/ how’s to interpret /discussion the data should be improved. Some of my suggestions are as follow:

P.S. Please add the line number throughout the text during revision stages.

ABSTRACT

- I suggest the outcome/result from the study should more in-depth discussion likes what microbial species/genes contributed/indicated to the spoilage of the fish and how? How’s those changes related with its meat quality (both texture and flavor/odor)? etc.

Response to reviewer: Special thanks to you for your good comments. According your comments, we check the results of function prediction (Section 3.6) and we found it is really true as you said that some gene functions have contributed to the meat quality. For example, we found the metabolism of terpenoids, polyketides and lips increased in C-O-D12, and those metabolisms could lead to decarboxylation, deamination and enzymatic oxidation. Decarboxylation, deamination and enzymatic oxidation will make the protein degrade and produce bad flavor, such as: mercaptan, H2S, indole. Meanwhile, studies proved that SSO can release alkaline protease to degrade myofibrillar protein which contribute to texture of crawfish.

Totally, we added those discussions in our new manuscript as follows:

Abstract: Revised, line 56-62.

Section 3.6: Rewrite, Line 400-414.

- Last sentence of the abstract: Author stated that the findings from this study can provide a theoretical guidance for the storage of crawfish at 4℃. How???? Based on microbial community changes??? Or what??? Please clarify this point.

Response to reviewer: We are very sorry for our mistakes. The goal of this study is to find the bacterial community changes and want to make contribution to the targeted bacteriostasis in our future study. Thus, we rewrite last sentence of abstract as follows: Those results are very important to achieve the purpose of targeted inhibition of crawfish spoilage under 4℃ with vacuum package. Line 61-62.

INTRODUCTION

- Some of review literature is not related to the work likes the paragraph of “Different packaging treatments”. However, instead of introduction, I think author should design to do experiment in all vacuum or MAP, compared with control. Unless, author should give the reason why select the vacuum for this study?

Response to reviewer: We really agreed your comments. MAP is a very good way to keep aquatic product fresh. While, crawfish is very small (50-100g and 50-90mm) and has high water content, so MAP is not cost-effective for crawfish, especially in market. Low temperature and vacuum packaging are the common methods for crawfish. According your suggestions, we deleted the “Different packaging…….” Line 102-106 and added the reason that why we choose refrigeration and vacuum packaging. Line 109-113.

MATERIAL AND METHODS

- I wonder why author use vacuum packaging for- this study? I had strong impact to the changes in both microbial population and quality, compared to traditional hand-wrapped or normal packaging. Why should author not design the experiment to compare between vacuum and control or etc.? (Moreover, I suggest the “vacuum packaging” should appear in the topic).

Response to reviewer:

1, why author use vacuum packaging for- this study?

Crawfish is high water content with high protein, refrigerated with vacuum packaging storage is more easily to keep quality and improve the shelf life of crawfish. Thus,   refrigerated with vacuum packaging is a common method to store crawfish[1], but the report about bacterial community changes and its relation to flavor/texture are very few.

[1] Lyon W J , Reddmann C S . Bacteria associated with processed crawfish and potential toxin production by Clostridium botulinum type E in vacuum-packaged and aerobically packaged crawfish tails. [J]. Journal of Food Protection, 2000, 63(12):1687.

2, Why should author not design the experiment to compare between vacuum and control or etc?

We really agreed your comments. It is more meaningful to compare the bacterial community changes between traditional hand-wrapped or normal packaging. However, other studies had already showed the results of bacterial community changes with traditional packaging under 4,5 and 25℃[2,3]. The reports about bacterial community changes of crawfish under 4℃ with vacuum packaging are very few. Moreover, different from other studies, we found Enterococcus aquimarinus was also a SSO when crawfish store at 4℃ with vacuum packaging. This finding is very meaningful for the purpose of targeted inhibition of crawfish spoilage under 4℃ with vacuum package. We added the discussion about comparing our results to other studies in revised manuscript. Line 325-333.

[2] Cox N A , Lovell R T .  Identification and characterization of the microflora and spoilage bacteria in freshwater crayfish Procambarus clarkii (Girard)[J]. Journal of Food Science, 2010, 38(4):679-681.

[3] Yan S , Yu D , Tang C , et al. Physicochemical and microbiological changes in postmortem crayfish (Procambarus clarkii) stored at 4°C and 25°C[J]. International Journal of Food Science & Technology.

3, Moreover, I suggest the “vacuum packaging” should appear in the topic.

Many thanks for your suggestion, we added vacuum packaging in our title.

- What is the slaughter process for this study since author start with the alive fish. How it proper for ethics or animal welfare? Please state the process in this part.

Response to reviewer: We are sorry for our mistake. The edible part of crawfish is mainly shrimp tail, so crawfish need be treated with slaughter. The slaughter process is : crawfish was cleaned in water, the head and sand vein were removed, the tail were rinsed. We added this in Line 125-128.

RESULT AND DISCUSSION

- I suggest author add more discussion to state the relationship between the changes in bacterial community and the changes in texture/flavor. This should be create the interesting data of this work.

Response to reviewer: Many thanks for your advises. We rewrite Section 3.6 and did more discussion about the relationship between the changes in bacterial community and the changes in texture/flavor. Please see Line 400-423. Moreover, we also combined the results of function prediction to discuss the relationship between bacterial function and flavor. Line 494-505.

P.S. reference is not appear in the correct format, please corrected it.

Response to reviewer: We are very sorry for our mistakes. We revised the format of reference in new manuscript.

Special thanks to you for your good comments.

We tried our best to improve the manuscript and made some changes in the manuscript. These changes will not influence the content and framework of the paper. And here we did not list the changes but marked in red in revised paper.

Reviewer 2 Report (Previous Reviewer 1)

The authors have not taken into account a large part of the recommendations that I made them observe in the first review. The discussion of results is very complex, difficult to understand, it is not well planned. There are important flaws, such as saying that in vacuum conditions grows significantly at 4º C for 12 days, an aerobic microorganism such as Enterococcus aquimarinus.

Author Response

Dear Editors and Reviewers:

Thank you for your letter and for the reviewers’ comments concerning our manuscript entitled “Bacterial community changes during freshwater-crawfish spoilage under the low- temperature storage” (ID: 1994477). Those comments are all valuable and very helpful for revising and improving our paper, as well as the important guiding significance to our researches. We have studied comments carefully and have made correction which we hope meet with approval. Revised portion are marked in red in the paper. The main corrections in the paper and the responds to the reviewer’s comments are as flowing:

R2:The authors have not taken into account a large part of the recommendations that I made them observe in the first review. The discussion of results is very complex, difficult to understand, it is not well planned. There are important flaws, such as saying that in vacuum conditions grows significantly at 4º C for 12 days, an aerobic microorganism such as Enterococcus aquimarinus.

Response to reviewer: We are very sorry that we didn’t carefully revise in our first round. This time we almost rewrote the part of Results and Discussion, as follows:

Section 3.1……. rewrite

Section 3.2……. rewrite

Section 3.3……. revise Line 269-275

Section 3.4……. revise Line 318-333

Section 3.5……. rewrite

Section 3.6……. rewrite

Section 3.7……. revise Line447-505

Conclusion……rewrite.

Moreover, strictly speaking, Enterococcus aquimarinus belongs to facultative bacteria [1].

[1] Michalik M . The Many Faces of Enterococcus spp.—Commensal, Probiotic and Opportunistic Pathogen[J]. Microorganisms, 2021, 9.

Many thanks for your suggestion and it has greatly improved our manuscript.

We tried our best to improve the manuscript and made some changes in the manuscript. These changes will not influence the content and framework of the paper. And here we did not list the changes but marked in red in revised paper.

Reviewer 3 Report (New Reviewer)

This article deals with the investigation on the bacterial community and some meat quality parameter changes of crawfish stored at 4℃.  The topic is interesting, the presentation of all data and their discussion are very good. Moreover, their results can provide strategies for keeping aquatic products fresh especially for crawfish. This manuscript still needs minor revision before publication, some of my suggestions are as follows:

Line 107, what do you mean by “MAP packaged prawns”??? is it Modified Atmosphere Packaging?? Please confirm.

Line 236, the abscissa of Figure.1 (a and b) is not the same. “Storage time” and “Time”..

Line 369, “4 ℃” deleted the space and changed as “℃”.

Line 239 is section 3.3 and Line 255 is also section 3.3, please confirm. 

Line 460, the format of Table.3 is not correct.

Please add the results of correlation of dominant bacteria and quality in section 4.

Author Response

Dear Editors and Reviewers:

Thank you for your letter and for the reviewers’ comments concerning our manuscript entitled “Bacterial community changes during freshwater-crawfish spoilage under the low- temperature storage” (ID: 1994477). Those comments are all valuable and very helpful for revising and improving our paper, as well as the important guiding significance to our researches. We have studied comments carefully and have made correction which we hope meet with approval. Revised portion are marked in red in the paper. The main corrections in the paper and the responds to the reviewer’s comments are as flowing:

R3:

This article deals with the investigation on the bacterial community and some meat quality parameter changes of crawfish stored at 4℃.  The topic is interesting, the presentation of all data and their discussion are very good. Moreover, their results can provide strategies for keeping aquatic products fresh especially for crawfish. This manuscript still needs minor revision before publication, some of my suggestions are as follow:

Line 107, what do you mean by “MAP packaged prawns”??? is it Modified Atmosphere Packaging?? Please confirm.

Response to reviewer: We are very sorry for our mistakes. According other reviewers, we deleted this sentence and added the reason that why we chose refrigeration and vacuum packaging.   

Line 236, the abscissa of Figure.1 (a and b) is not the same. “Storage time” and “Time”..

Response to reviewer: We are very sorry for our mistakes. We replaced Figuire.1b in our new manuscript.

Line 369, “4 ℃” deleted the space and changed as “℃”.

Response to reviewer: We are very sorry for our mistakes. We revised this in line 350.

Line 239 is section 3.3 and Line 255 is also section 3.3, please confirm.

Response to reviewer: We are very sorry for our mistakes. We reorder the part of Results and Discussions.

Line 460, the format of Table.3 is not correct.

Response to reviewer: We are very sorry for our mistakes. We revised the format of Table.3.

Pleas add the results of correlation of dominant bacteria and quality in section 4.

Response to reviewer: We are very sorry for our mistakes. We rewrite the Conclusion and added the results of correlation of dominant bacteria and quality.

Special thanks to you for your good comments.

We tried our best to improve the manuscript and made some changes in the manuscript. These changes will not influence the content and framework of the paper. And here we did not list the changes but marked in red in revised paper.

Round 2

Reviewer 1 Report (Previous Reviewer 2)

The authors conducted the required revisions as suggested before.

Thus, the quality of this study is fair.

However, I think the weak point of this work is the originally design of experiment. It lacks of experiment set to the obvious goal/findings.

The MS needs to be largely re-worked. The findings of the study are fairly discussed. Also, the scientific soundness is low.

Reviewer 2 Report (Previous Reviewer 1)

I thank the authors for the changes made to the manuscript according to the recommendations I gave them.

These changes have considerably improved the scientific quality of the writing.

This manuscript is a resubmission of an earlier submission. The following is a list of the peer review reports and author responses from that submission.

Round 1

Reviewer 1 Report

Figure 1, (a). It puts -4ºC and it should put 4ºC. We are talking about refrigeration temperatures above 0ºC.

In that same Figure 1 (b), the same as in the text, my suggestion is that they talk about total aerobic microorganisms and not TPC. The TPC is the culture medium used for counting total aerobic microorganisms.

I do not understand what the authors are aiming for in exploring SSO in crayfish, stored at 4℃, by analyzing microbiota profiles based on 16S rDNA length in conjunction with High-Through Sequencing Analysis. Why have they not analyzed how these microorganisms evolve at room temperature? ambient? What would be interesting would be to see the evolution experienced by microbial species with high resistance to cold and with great decomposition capacity, as is the case of Aeromonas sobria and Shewanella putrefaciens. I consider it more interesting to have related the Total volatile base nitrogen (TVB-N) and Texture profile analysis (TPA) with the microbial evolution at 4ºC and 25ºC of these aerobic microorganisms, taking into account the great decomposition capacity that both bacterial species have even at refrigeration temperatures.

According to the authors and how one would expect (lines 250-264) the richness of aerobic bacteria such as: Paucibacter oligotrophus; Cloacibacterium rupense, etc., has decreased (see Table 1). However, I do not understand how in Figure 1(b), the results of the TPC increase as the storage time increases, especially at 4ºC. With the TPC we count microorganisms total aerobes. In vacuo, the results of the TPC presented by the authors are not consistent.

In Figure 3 (Crayfish Flavor Volatile Compounds under 25℃ and 4℃) aldehydes appear. However, in Figure 6 (Correlation heat map of dominant bacteria and volatile flavor compounds.) aldehydes are not included in that correlation. My question to the authors is directed towards the omission in Figure 6 of the aldehydes.

Author Response

Dear Reviewers:

Thank you for your letter and for the reviewers’ comments concerning our manuscript entitled “Bacterial community changes during freshwater-crawfish spoilage under the low- temperature storage” (ID: 1870877). Those comments are all valuable and very helpful for revising and improving our paper, as well as the important guiding significance to our researches. We have studied comments carefully and have made correction which we hope meet with approval. Revised portion are marked in red in the paper. The main corrections in the paper and the responds to the reviewer’s comments are as flowing:

Figure 1, (a). It puts -4ºC and it should put 4ºC. We are talking about refrigeration temperatures above 0ºC.

Response to reviewer: We are very sorry for our mistakes, we revised this. It should be 4℃.

In that same Figure 1 (b), the same as in the text, my suggestion is that they talk about total aerobic microorganisms and not TPC. The TPC is the culture medium used for counting total aerobic microorganisms.

Response to reviewer: We are very sorry for our mistakes, it really true that TPC is the culture medium used for counting total aerobic microorganisms. We changed TPC as total aerobic microorganisms.

I do not understand what the authors are aiming for in exploring SSO in crayfish, stored at 4℃, by analyzing microbiota profiles based on 16S rDNA length in conjunction with High-Through Sequencing Analysis. Why have they not analyzed how these microorganisms evolve at room temperature? ambient? What would be interesting would be to see the evolution experienced by microbial species with high resistance to cold and with great decomposition capacity, as is the case of Aeromonas sobria and Shewanella putrefaciens. I consider it more interesting to have related the Total volatile base nitrogen (TVB-N) and Texture profile analysis (TPA) with the microbial evolution at 4ºC and 25ºC of these aerobic microorganisms, taking into account the great decomposition capacity that both bacterial species have even at refrigeration temperatures.

Response to reviewer: We are very sorry for our mistakes, the responses are as follows:

  • I do not understand what the authors are aiming for in exploring SSO in crayfish, stored at 4℃, by analyzing microbiota profiles based on 16S rDNA length in conjunction with High-Through Sequencing Analysis

Those results can provide guidance for the targeted antibacterial (our future researches). We revised this in our new abstract.

  • Why have they not analyzed how these microorganisms evolve at room temperature?

According your comments and other reviewers, we deleted the part of 25℃, since the works of this paper mainly included:1) the quality changes of crawfish stored at 4℃, such as: TVB-N, TPC, texture and volatile flavor compounds; 2) its bacterial changes during the storage. And we wanted to reflected the structure (species level) and composition of microbial community in crawfish during storage at 4℃; 3) the relationship between quality (flavor) changes and bacterial community.

 The reason that we did not analysis room temperature is the shelf life of crawfish is only 2 days under 25℃, but it can be prolonger to 12 days under 4℃. Moreover, most of the aquatic product was stored at 4℃. And our future mainly focus on the targeted antibacterial for crawfish stored at 4℃.

  • I consider it more interesting to have related the Total volatile base nitrogen (TVB-N) and Texture profile analysis (TPA) with the microbial evolution at 4ºC and 25ºC of these aerobic microorganisms

We really agreed with your comments that it is meaningful to have related the Total volatile base nitrogen (TVB-N) and Texture profile analysis (TPA) with the microbial evolution at 4ºC of these aerobic microorganisms. Thus, we added the correlation analysis between TVB-N, TPA with microbial.

According to the authors and how one would expect (lines 250-264) the richness of aerobic bacteria such as: Paucibacter oligotrophus; Cloacibacterium rupense, etc., has decreased (see Table 1). However, I do not understand how in Figure 1(b), the results of the TPC increase as the storage time increases, especially at 4ºC. With the TPC we count microorganisms total aerobes. In vacuo, the results of the TPC presented by the authors are not consistent.

     Response to reviewer: Thank you for your comments, we really agreed with your suggestion that the aerobic bacterial decreased in this study since in vacuo. While, our results reflected that Aeromonas sobria; Shewanella putrefaciens; Trichococcus pasteurii; and Enterococcus aquimarinus were the specific spoilage organisms and those bacterial belonged to facultative anaerobic bacteria [1,2,3,4] which means they can survive in both aerobic and anaerobic conditions. That is the reason that total aerobes count increased and we added this explanations in Line 262-264.

[1] Ha A , Mll A , Gw B . Complete genome sequence data of multidrug-resistant Aeromonas veronii strain MS-18-37 - ScienceDirect[J]. Data in Brief, 23.

[2] Removal of Shewanella putrefaciens Biofilm by acidic electrolyzed water on food contact surfaces

[3] A W H C , A Y T L , B J G L , et al. Facultative-like anaerobic packed bed reactor treating low strength wastewater: microbial community and energy balance appraisements. 2021.

[4]  Michalik M . The Many Faces of Enterococcus spp.—Commensal, Probiotic and Opportunistic Pathogen[J]. Microorganisms, 2021, 9.

In Figure 3 (Crayfish Flavor Volatile Compounds under 25℃ and 4℃) aldehydes appear. However, in Figure 6 (Correlation heat map of dominant bacteria and volatile flavor compounds.) aldehydes are not included in that correlation. My question to the authors is directed towards the omission in Figure 6 of the aldehydes.

Response to reviewer: We are very sorry for our mistakes, we revised Figure.6 and added the aldehydes.

Special thanks to you for your good comments.

We tried our best to improve the manuscript and made some changes in the manuscript. These changes will not influence the content and framework of the paper. And here we did not list the changes but marked in red in revised paper.

We appreciate for Reviewers’ warm work earnestly, and hope that the correction will meet with approval.

Once again, thank you very much for your comments and suggestions.

Reviewer 2 Report

This article deals with the investigation on the bacterial community and some meat quality parameter changes of crawfish stored at 4 and 25℃.  Actually, the topic is interesting, but I think the presentation of all data (as well as their discussion) is not good enough. The text is too long, and it is difficult to understand/follow the discussion. Some of my suggestions are as follow:

- The topic is bacterial changes during low-temperature storage, but the temperature used in this study is 4℃ and room temperature (25℃)????

ABSTRACT

- Line 14: The crawfish is hard to preserve because of its high content of protein???? I do not think so since the protein content is not the main reason to cause fish spoilage. Please check the accuracy or re-write this sentence.

- Line14-16: Author stated that this study aimed to determine the changes in TVB-N, TPC, texture, volatile of crawfish during storage? This is not corresponding with the topic. Also, how about the monitoring the bacterial community changes???? I suggested to check and re-write the abstract for clarifying.

- Line17-18: If these sentences are the findings from this study. It is too general and did not gain new data/info. or any novel knowledge. I suggested that author cannot conclude that the preservation temp. was a critical factor since this study did not measure/monitor the spoilage as affected by many factors. Author mentioned that “lower temperature could support a longer shelf-life”--- This is also basic. Therefore, I suggested author to re-write this MS to show out the interested info. related with the changes in bac. community and other meat quality as affected by low temp.

Author Response

Dear Reviewers:

Thank you for your letter and for the reviewers’ comments concerning our manuscript entitled “Bacterial community changes during freshwater-crawfish spoilage under the low- temperature storage” (ID: 1870877). Those comments are all valuable and very helpful for revising and improving our paper, as well as the important guiding significance to our researches. We have studied comments carefully and have made correction which we hope meet with approval. Revised portion are marked in red in the paper. The main corrections in the paper and the responds to the reviewer’s comments are as flowing:

This article deals with the investigation on the bacterial community and some meat quality parameter changes of crawfish stored at 4 and 25℃.  Actually, the topic is interesting, but I think the presentation of all data (as well as their discussion) is not good enough. The text is too long, and it is difficult to understand/follow the discussion. Some of my suggestions are as follow:

Response to reviewer: Special thanks to you for your good comments, we tried our best to improve our manuscript according your comments. According your comments and other reviewers, we decided to delete the part of 25℃ since this manuscript mainly focus on the quality and bacterial community changes of crawfish which stored at 4℃.

- The topic is bacterial changes during low-temperature storage, but the temperature used in this study is 4℃ and room temperature (25℃)????

Response to reviewer: We are very sorry for our mistakes, the works of this paper mainly included:1) the quality changes of crawfish stored at different temperature such as: TVB-N, TPC, texture and volatile flavor compounds; 2) its bacterial changes during the storage. And we wanted to reflected:1) the structure (species level) and composition of microbial community in crawfish during storage at 4℃; 3) the relationship between quality (flavor) changes and bacterial community. According your comments, we changed the title as: The quality and bacterial community changes during freshwater-crawfish stored at 4℃.

ABSTRACT

- Line 14: The crawfish is hard to preserve because of its high content of protein???? I do not think so since the protein content is not the main reason to cause fish spoilage. Please check the accuracy or re-write this sentence.

Response to reviewer: We are very sorry for our mistakes, we re-write the abstract. This sentence changed to: Crawfish can be easily to spoiled due to the rich nutrition and content of high water, which is difficult to preserve.

- Line14-16: Author stated that this study aimed to determine the changes in TVB-N, TPC, texture, volatile of crawfish during storage? This is not corresponding with the topic. Also, how about the monitoring the bacterial community changes???? I suggested to check and re-write the abstract for clarifying.

Response to reviewer: We are very sorry for our mistakes, we re-write our results. Those sentence were as follows: at 4℃, the specific spoilage organisms (species level) of crawfish were: Aeromonas sobria; Shewanella putrefaciens; Trichococcus pasteurii; and Enterococcus aquimarinus since their abundances significantly increased after stored for 12 days at 4℃…… Line 79-81.

- Line17-18: If these sentences are the findings from this study. It is too general and did not gain new data/info. or any novel knowledge. I suggested that author cannot conclude that the preservation temp. was a critical factor since this study did not measure/monitor the spoilage as affected by many factors. Author mentioned that “lower temperature could support a longer shelf-life”--- This is also basic. Therefore, I suggested author to re-write this MS to show out the interested info. related with the changes in bac. community and other meat quality as affected by low temp.

Response to reviewer: We are very sorry for our mistakes, according your meaningful comments, we re-write abstract. Line 74-91.

Special thanks to you for your good comments.

We tried our best to improve the manuscript and made some changes in the manuscript. These changes will not influence the content and framework of the paper. And here we did not list the changes but marked in red in revised paper.

We appreciate for Reviewers’ warm work earnestly, and hope that the correction will meet with approval.

Once again, thank you very much for your comments and suggestions.

Reviewer 3 Report

The author studied the bacterial community changes in freshwater-crawfish spoilage at 4℃ and 25℃, and different indexes, such as total volatile base nitrogen (TVB-N), total plate count (TPC), texture and volatile flavor compounds, were used to determine the quality of crawfish. However, the experiment design and concept is so similar with the previous study. I think the novelty of this study is lack and cannot recommend the manuscript for publication to this journal due to the following reasons:

1. In the manuscript, Line 48, Please delete the extra letter b.

2. Line 54, please delete the extra space.

3. Line 55, please change the word “spoil” to “spoilage”.

4. Line 107, please delete the extra letter a (p<0.05).

5. Line 178, please change the sentence “control he increasing of TVB-N value of the samples” to “inhibit the increase of TVB-N value of samples”.

6. Line 189, please change “on” to “from”.

7. Line 200, please change “late” to “later”.

8. Figure 2c, please change the abscissa unit from “time” to “day”, and change “time” to “storage time” in other pictures.

9. The title of this manuscript is “Bacterial community changes during freshwater-crawfish spoilage under the low- temperature storage”, but how to evaluate the correlation of TVB-N, TPC, texture and volatile flavor compounds?

10. Line238, the author mentioned that 4℃ and 25℃ group were set as the targets to analyze changes of microbial diversity, but Table 1 only showed the abundance of species for crawfish stored at 4 ℃.

11. The changes in the microbial diversity of crayfish during storage are also important, while the authors only list the initial and final microflora.

Author Response

Dear Reviewers:

Thank you for your letter and for the reviewers’ comments concerning our manuscript entitled “Bacterial community changes during freshwater-crawfish spoilage under the low- temperature storage” (ID: 1870877). Those comments are all valuable and very helpful for revising and improving our paper, as well as the important guiding significance to our researches. We have studied comments carefully and have made correction which we hope meet with approval. Revised portion are marked in red in the paper. The main corrections in the paper and the responds to the reviewer’s comments are as flowing:

The author studied the bacterial community changes in freshwater-crawfish spoilage at 4℃ and 25℃, and different indexes, such as total volatile base nitrogen (TVB-N), total plate count (TPC), texture and volatile flavor compounds, were used to determine the quality of crawfish. However, the experiment design and concept is so similar with the previous study. I think the novelty of this study is lack and cannot recommend the manuscript for publication to this journal due to the following reasons:

Response to reviewer: Special thanks to you for your good comments, we tried our best to improve our manuscript according your comments. In this manuscript, the works of this paper mainly included:1) the quality changes of crawfish stored at 4℃ such as: TVB-N, TPC, texture and volatile flavor compounds; 2) its bacterial changes during the storage. And we wanted to reflected:1) the structure (species level) and composition of microbial community in crawfish during storage at 4℃; 3) the relationship between quality changes (flavor) and bacterial community. Moreover, according your comments and other reviewers, we decided to delete the part of 25℃ since this manuscript mainly focus on the quality and bacterial community changes of crawfish which stored at 4℃.

Regarding the uniqueness of the study, we think that the test indications we have selected are common indicators in the study of meat or aquatic goods. Although many studies have focused on the changes in microbial composition that occur during the preservation of aquatic products, studies in microorganism are still primarily concerned with describing particular microorganisms (such as DGGE) or the horizontal changes in microbial genera through second generation sequencing. To ensure the reproducibility of the community and lessen the impact of total DNA extraction on sequencing samples, the application of third-generation high-throughput sequencing technology in complex environments such as food requires needs to be optimized in sample pretreatment and microbial enrichment. The change in the microbial community of aquatic products is studied in this paper using third-generation high-throughput sequencing technology for the first time. This study reflects the change in the microbial community at the species level and demonstrates the viability of third-generation high-throughput sequencing in the study of food preservation and corruption. However, the contribution of sequencing technology to the novelty of this study has not been best reflected. The correlation analysis between sequencing data and different indicators is weakened as a result of our flaws in the time point selection for the sequencing, which makes the contribution of sequencing technology to the innovation of this paper less well reflected. Nevertheless, we still hope to offer some suggestions for further research in this area.

  1. In the manuscript, Line 48, Please delete the extra letter b.

Response to reviewer: We are very sorry for our mistakes, according your comments and other reviewers, we re-write the abstract. Line 74-91.

  1. Line 54, please delete the extra space.

Response to reviewer: We are very sorry for our mistakes, we deleted the extra space. Line 124.

  1. Line 55, please change the word “spoil” to “spoilage”.

Response to reviewer: We are very sorry for our mistakes, we changed this word. Line 125.

  1. Line 107, please delete the extra letter a (p<0.05).

Response to reviewer: We are very sorry for our mistakes, we deleted “a”. Line 238.

  1. Line 178, please change the sentence “control he increasing of TVB-N value of the samples”

to “inhibit the increase of TVB-N value of samples”.

Response to reviewer: We are very sorry for our mistakes, we changed this sentence. “Those results indicated that low temperature could inhibit the increase of TVB-N value of the samples during storage. ” Line 249-250.

  1. Line 189, please change “on” to “from”.

Response to reviewer: We are very sorry for our mistakes, we revised this in Line 261.

  1. Line 200, please change “late” to “later”.

Response to reviewer: We are very sorry for our mistakes, we revised this in Line 275.

  1. Figure 2c, please change the abscissa unit from “time” to “day”, and change “time” to “storage time” in other pictures.

Response to reviewer: We are very sorry for our mistakes, we revised this in Figure 2 and Figure 1.

  1. The title of this manuscript is “Bacterial community changes during freshwater-crawfish spoilage under the low- temperature storage”, but how to evaluate the correlation of TVB-N, TPC, texture and volatile flavor compounds?

Response to reviewer: According your comments, we changed the title of this manuscript. “The quality and bacterial community changes during freshwater-crawfish stored at 4℃”. We also found that the unpleasant flavor of crawfish was mainly positively to specific spoilage organisms, such as: Alkyl phenol polyoxyethylene ether (2-(Ethenyloxy)-ethanol) and S-containing (Dimethyl trisulfide and Tetrasulfide, dimethyl) were significantly positively correlated Enterococcus, Trichococcus, Shewanella and Aeromonas. Part 3.5.

  1. Line238, the author mentioned that 4℃ and 25℃ group were set as the targets to analyze changes of microbial diversity, but Table 1 only showed the abundance of species for crawfish stored at 4 ℃.

     Response to reviewer: We are very sorry for our mistakes, we deleted this sentence. This manuscript mainly focused on 4℃, since there are few reports on the quality and bacterial community changes during freshwater-crawfish stored at 4℃.

  1. The changes in the microbial diversity of crayfish during storage are also important, while the authors only list the initial and final microflora.

   Response to reviewer: It is really true that microbial diversity of crawfish during storage are important. This article mainly focused on finding the dominant spoilage bacteria of crayfish and their quality changes under 4ºC, and those results can provide guidance for the targeted antibacterial (our future researches). We will make further analysis in future research about the microbial diversity of crawfish during storage.

Special thanks to you for your good comments.

We tried our best to improve the manuscript and made some changes in the manuscript. These changes will not influence the content and framework of the paper. And here we did not list the changes but marked in red in revised paper.

We appreciate for Reviewers’ warm work earnestly, and hope that the correction will meet with approval.

Once again, thank you very much for your comments and suggestions.

Round 2

Reviewer 1 Report

Line 436: "In this study, the crawfish was stored at different temperatures". This sentence should not appear. Finally, the study was only carried out at 4ºC.

Figure 1 (line 90). Instead of TPC put "count of aerobic microorganisms"

Line 74: replace total plate count (TPC) with total aerobic microorganisms

The samples were vacuum packed (line 81). However, throughout the experiment, they do not take into account the influence that this packaging system exerts, together with the cold, on the final microbial composition. Can you explain to me why you omit this factor? so important to conservation?

Throughout the text there are errors of form: sometimes they write Figure and others Fig. Check these forms of writing carefully. Sometimes they highlight them in bold and other times, they do not.

Reviewer 2 Report

Dear, author

 This revised version is not significantly improved as per I expect. Structure/research design, presentation of data as well as the discussion are hard to catch up the contents. I have tried to give some suggestions, I suggested author to develop these points throughout the MS, but author only revised their abstract.

- Thus, I think this version may not appropriate for publication. However, this original data/findings of this MS had the potential to improve/re-construct for further opportunity to public elsewhere.

Thanks

Thanks

Reviewer 3 Report

The manuscript can be accepted in this form.